# Institutional factors influencing knowledge production for practice: Evidence from nonprofit studies

**Ji Ma** [1]*, **Joycelyn Ovalle**[1], **Yan Wang**[2]

**1** The LBJ School of Public Affairs and RGK Center, The University of Texas at Austin, Austin, TX, United States of America, **2** Department of Sociology, Zhejiang University, Hangzhou, Zhejiang, China

* maji@austin.utexas.edu

**Data Availability Statement:** All datasets are available via OSF (https://osf.io/qj85m/).

**Funding:** The project is partly funded or supported by (1) the Academic Development Funds from the RGK Center, (2) a 2021-22 PRI Award from the

## Abstract

This study theorizes and tests an institutional-logics framework to explain why some universities produce more practice-oriented peer-reviewed journal articles than others, using nonprofit studies as an example. Empirically, knowledge production for practice can be increased by *(1)* graduate degree programs with an emphasis on nonprofit management, *(2)* research centers on nonprofit studies, and *(3)* location in disadvantaged communities; however, *(4)* status as an R1 or R2 research university substantially decreases the production of practical knowledge. Furthermore, *(5)* research centers can mediate the influence of community needs on knowledge production, so that universities with nonprofit research centers are more responsive to solving community issues. Theoretically, knowledge production follows the institutional logics of both closed and open systems, and institutions such as research centers that can repackage the culture of open systems to make it acceptable to closed systems are essential mediators.

## Introduction

Practical scholarship offers critical knowledge for understanding and solving "real-world" problems. However, such efforts have been a difficult endeavor in many research fields of social sciences, such as sociology, political science, management, and implementation science. The gap between theoretical and practical knowledge can be interpreted through three lenses: knowledge transfer problems, the nature-of-knowledge problem, and knowledge production problems [1]. Applying research findings to practice requires additional efforts of transformation, which often do not align with the interests of scholars and are not as highly valued within academia compared to theoretical works. Theoretical knowledge and practical knowledge also differ epistemologically because they are different ways of knowing. The former is believed to unveil the nature of things, context-free, and to be committed to building generalizable laws and principles. While the latter is often context-specific, made for solving particular problems, limited in generalizability, and treated as derivative from theory and therefore a secondary way of knowing.

LBJ School, (3) library resources through the IU Lilly Family School of Philanthropy, and (4) cloud computing resources through the Texas Advanced Computing Center at UT Austin. The funders had no role in study design, data collection, analysis, decision to publish, or preparation of the manuscript.

**Competing interests:** The authors have declared that no competing interests exist.

The gap between theory and practice, itself theoretically a knowledge production problem, becomes an institutional problem in practice. The academic tenure process, funding criteria, and editorial policies largely favor theory-building but discount practical knowledge. This viewpoint has attracted growing attention because it targets the root causes and provides potential routes to change. Take implementation science for example. This research field was initially created to systematize the implementation of evidence-based programs, highlights the importance of institutional contexts that can impede the integration of evidence or practice-based programs or initiatives in organizations [2, 3]. However, merely having these practices available is not enough for widespread adoption and successful implementation. It is crucial to consider the institutional contexts in which these practices are intended to be implemented, taking into account factors such as leadership support, organizational culture, resource availability, workforce capacity, and external influences [4].

This study adopts the knowledge production perspective and uses nonprofit studies, a social science research field tightly connected to practice, as an example to examine the institutional factors that influence the production of practical knowledge. We studied these factors at three levels of analysis: *program*, *organization*, and *community*. We also theorize a framework of institutional logics to interpret how these factors influence knowledge production for practice. We empirically test this framework with data from various sources, state-of-the-art (SOTA) natural language processing (NLP) algorithms, and robust statistical analysis. In a nutshell, in order to produce more practice-oriented scholarship, the knowledge production process needs to be facilitated by an open-system logic that emphasizes the engagement of various stakeholders, such as students and local communities. The process can also be mediated through knowledge transfer channels, such as research centers on nonprofits and philanthropy, that can shift the logics between closed and open systems and facilitate the sharing of knowledge between scholars and practitioners.

## Narrowing the divide: In search of a coherent framework

Nonprofit studies is an interdisciplinary social science research field that focuses on the practices of nonprofit organizations and originated in the 1920s [5]. The practical needs of the nonprofit sector and policy making are key to the existence of this field and related educational programs [6, 7], but the gap between scholars and practitioners has a history as old as the field itself. Scholars found that although academic publications on nonprofit studies in the field's early years responded to practitioners' common concerns, the coverage of these articles was much narrower than practitioners' needs [8, 9]. A review of a leading journal of nonprofit studies, for example, revealed that only 23 percent of the journal's publications between 2000 and 2010 distilled practical implications for nonprofits [10]. Such a pattern underscores the point that academica's concerns are predominantly theoretical and often deviate from those of practitioners [8]. Unfortunately, that situation has persisted for decades in this research field [11].

To narrow the gap, communication and collaboration between scholars and practitioners—for example, engaging practitioners at different stages of research, such as defining research questions, sharing data, and peer review [8, p. 306]—has been a key focus and the solution most referred to. The participation of practitioners in research can also influence academic research and teaching in broader ways, such as identifying new areas of research and changing teaching practices [11, p. 99]. [12] synthesized four important considerations for a successful collaboration: (1) relational, emphasizing building personal ties through different venues; (2) philosophical, recognizing the difference between different professions' norms; (3) organizational, underlining the organizational culture and needs behind collaboration; and (4) political

and ethical, such as disciplinary politics and directing research focus toward academic benefits.

As the four considerations for a successful collaboration suggest, the gap between theoretical and practical knowledge is intricate. In the research field of nonprofits and philanthropy, scholars also explored other possibilities. For example, [13]. suggested that the theory-practice gap can be narrowed by adopting an "abduction" process. This epistemological approach is different from conventional deduction and induction approaches in its emphasis on "[c]reating a testable [and creative] hypothesis that best explains the surprising phenomenon" [13, p. 209]. The rapid growth of degree programs in higher education can also bridge the theory-practice divide [7, 14–16]. Nonprofit centers, by facilitating collaborations and supporting both scholars and practitioners, hold especially important roles in the knowledge transfer and knowledge production process [17].

The reasons for the theory-practice divide are multifaceted, and the scholarly bridging efforts as well as the perspectives informing them are scattered. We are in dire need of a coherent framework to interpret the mechanisms through which different factors can influence knowledge production for practice.

## Producing practical knowledge: Influential factors at three levels

Existing studies on narrowing the theory-practice divide and related topics suggest that the analysis of influential factors works at three levels: the *program* level, which investigates the relationship between knowledge production for practice and university degree programs; the *organization* level, which focuses on organizational features such as the roles played by research centers on nonprofits and philanthropy; and the *community* level, which examines the connections between knowledge production and its external environment.

**Program-level factors.** In response to the rapid growth of the nonprofit sector in the United States and worldwide, degree programs in nonprofit and philanthropic studies expanded widely between the mid-1990s and the mid-2010s [18, 19]. The success of these programs depended on how effectively they equipped students with practical skills [20], especially for graduate students who came from or were expected to work in nonprofits [21]. To meet this expectation, nonprofit scholars were incentivized to produce more practical knowledge, even though other forces such as academic recognition and promotion standards pulled their attention toward theory building.

**Organization-level factors.** The organizational features favorable or unfavorable to practical knowledge production form another stream of factors. For example, universities with research centers focused on studying nonprofits and philanthropy are more connected to practitioners because these centers serve as crucial hubs for facilitating scholar-practitioner collaboration and knowledge transfer [17, 22]. On the negative side, the university tenure system and publication processes of many peer-reviewed journals are known as unfavorable to practical knowledge because their purposes are primarily academic and oriented toward theory-building [1, 23].

**Community-level factors.** Organizations are embedded in and influenced by their external environment, drawing personnel and resources from outside. From this perspective, organizations can be seen as "open systems" [24, pp. 87–106]. To be successful, an organization needs to assess and respond effectively to its external environment. In higher education, the success of a university also depends in part on how it responds to the needs of its external communities (e.g., initiatives aimed at reducing the local poverty rate of the institution's vicinity) [25]. Universities can respond to and engage their external neighborhoods through numerous

channels, such as communal participatory research [26] and service learning through curricular activities [27–29].

## Connecting the dots: A framework of institutional logics for producing practical knowledge

**Open- and closed-system logics.** From an institutional perspective, the process of producing practical knowledge is influenced by a complex variety of factors, ranging from degree programs to environmental contexts. In this section, we propose a coherent framework from an institutional-logics perspective to explain how these factors work together to influence the knowledge production process.

In organizational sociology, institutional logics are "systems of cultural elements (values, beliefs, and normative expectations) by which people, groups, and organizations make sense of and evaluate their everyday activities, and organize those activities in time and space" [30, p. 1]. The specific definition of institutional logics varies by theorist, but the central focus of this concept is to interpret individual and organizational behaviors within a social and institutional context; the institutional context both regularizes behavior and offers opportunities to change [31, p. 102].

Drawing from literature on institutional logics and higher education [32], identified two central logics in the field of U.S. higher education: open-system and closed-system. The *open-system* logic treats universities and research disciplines as open systems in their engagement of various stakeholders (e.g., students, teachers, and practitioners) in learning and knowledge production. The *closed-system* logic assumes universities and research disciplines as "storehouses of knowledge" and emphasizes the authority of academic institutions and faculty in teaching and research. The U.S. higher education experienced a major shift of institutional logic from the closed-system to the open-system type after World War II [32, pp. 322–323]. Academic disciplines can also vary in terms of "openness." For example, the applied social sciences usually adopt an open-system logic and usually respond to practical needs faster than theoretical social sciences [33].

By analyzing the factors influencing knowledge production, it is evident that all the conditions favorable for producing practical knowledge follow an open-system logic. They all facilitate exchange between academia and various outside stakeholders. For example, the degree programs emphasizing practical knowledge prepare graduate students for their professions, nonprofit research centers are founded to facilitate collaboration between scholars and practitioners, and universities friendly to practical knowledge respond to community issues through research and curricular activities.

By contrast, the *closed-system logic* emphasizes the authoritative roles of universities in knowledge production and focuses on academic interests that may not align with the interests of practitioners. For example, the primary readers of academic publications are scholars. Many of the research topics (e.g., overall structure of the nonprofit sector and methodology) help only with theory-building but not with problem-solving [10, p. 508]. Universities are evaluated and ranked by peers according to their academic performance, which largely depends on factors unaligned with the interests of practitioners, such as publications, research grants, and citations. The closed-system logic pulls universities towards serving academic interests and away from producing practical knowledge.

Combining the influencing factors and institutional logics, Table 1 summarizes our research framework and hypotheses. At each of the levels (program, organization, and community), the influencing factors can affect the production of practical knowledge through either of two institutional logics: the open-system logic, which connects universities to their

**Table 1. Producing practical knowledge: An institutional-logics framework.**

| | | Institutional logics of universities | | Operationalization |
|---|---|---|---|---|
| | | **Open-system** | **Closed-system** | |
| Influencing factors | Program | Train students to solve practical problems | Train students to comprehend and build theories | Graduate program (+) |
| | Organization | Facilitate collaboration and knowledge transfer | Serve academic system and tenure process | Nonprofit research centers (+), Research universities (-) |
| | Community | Respond to community needs | Operate as academic ivory towers | Poverty rate at institutional location (+) |

external environments, or the closed-system logic, which insulates higher education and research from responding to practical needs.

**Mediating role of research centers: Shifting logics between community and academia.** Producing knowledge in the form of academic publications is a primary function of academia. How, then, can the demands of external communities be internalized by academic institutions? Researchers suggest the need for channels for "shifting logics" and "cultural repackaging" (repackaging the culture of one system to be accepted by the other) in higher education [32]. As many studies have suggested, nonprofit research centers have been central players in translating practical needs into academic research [22, 34–37]. Fig 1 illustrates the possible logic-shifting role of nonprofit research centers: these centers can act as mediators enhancing the influence of community needs on knowledge production for practice.

## Testing theorization

Among the U.S. universities with an emphasis on nonprofit management education, why some universities produce more practical knowledge on nonprofits and philanthropy than others? We draw the following hypotheses from the framework (i.e., Table 1) to respond to this question.

**Hypothesis 1**: Universities with *graduate programs* in nonprofit and philanthropic studies produce more practice-oriented scholarship because these programs prepare students for their professions and introduce more practice-oriented topics to faculty research agendas.

**Hypothesis 2**: *Research universities* produce less practical knowledge because of their emphasis on theory-building.

**Hypothesis 3**: Universities with *nonprofit research centers* produce more practical knowledge because these centers can facilitate the exchange between scholars and practitioners.

**Hypothesis 4**: Universities located in counties with a high *poverty rate* produce more practical knowledge on nonprofits and philanthropy because these communities demand more practical solutions.

We draw the following hypothesis of interaction to test the mediating role of nonprofit research centers illustrated in Fig 1.

**Hypothesis 5**: Nonprofit research centers can *mediate* the relationship between poverty rate and practical knowledge: For universities with nonprofit research centers, the influence of poverty rate on producing practical knowledge is more substantial than for those without.

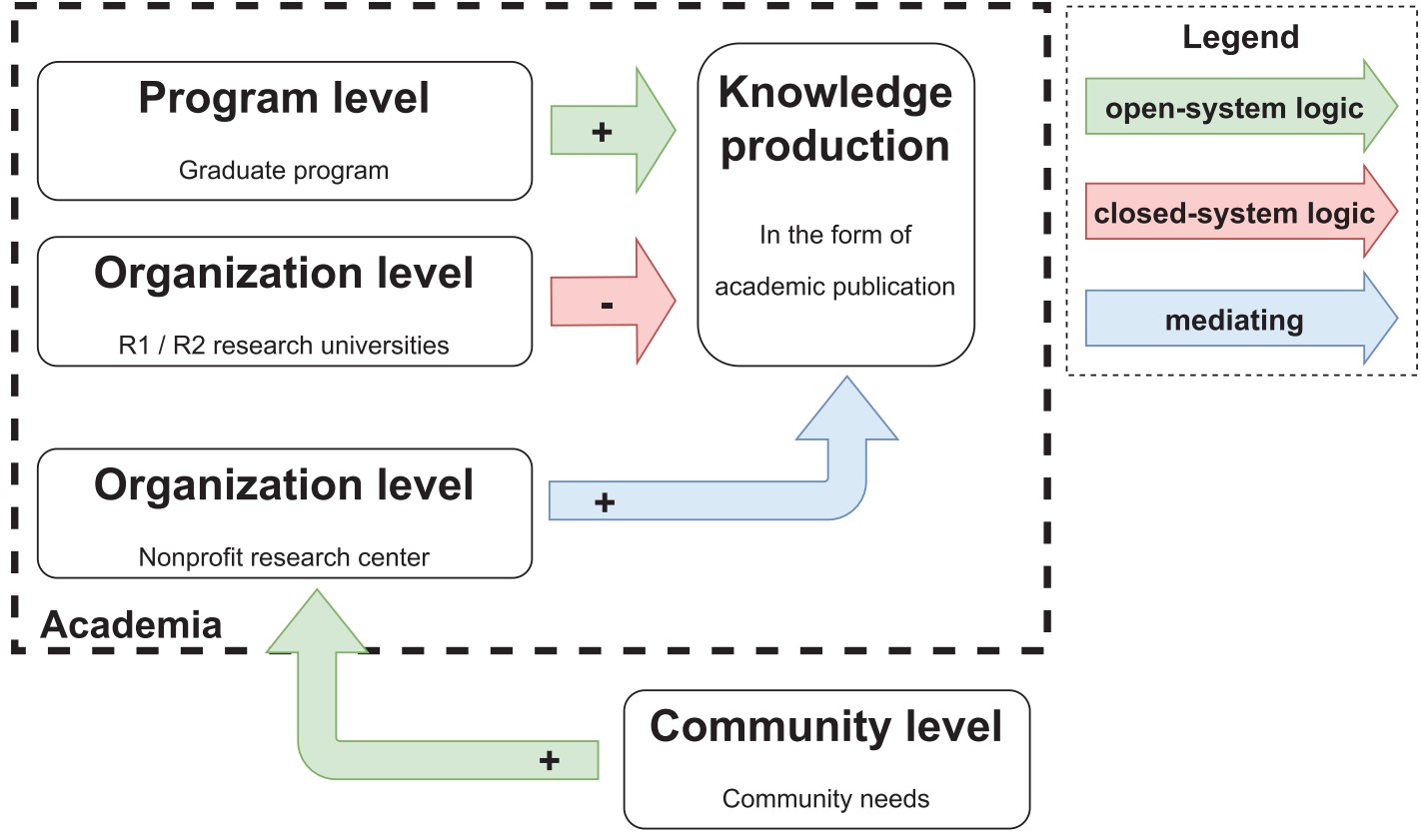

**Fig 1. Mediating role of research centers: Shifting logics from community to academia.**

## Materials and methods

### Data

We constructed datasets on U.S. universities and their publications on nonprofits and philanthropy from the following sources:

1. *Nonprofit Management Education, Current Offerings in University-Based Programs (NME)* is a well-known database developed and maintained by Mirabella et al. [18]. The NME documents the details of all degree and non-degree programs in nonprofit studies offered by U.S. universities. We selected the universities and programs for analysis using this database. There are 401 lines of records from this source, and each line records a department, school, or university with a nonprofit program (degree or non-degree).

2. A list of *Nonprofit Research Centers (NRC)* is compiled from numerous articles introducing the research centers studying nonprofits and philanthropy and located in U.S. universities [17, 18, 22, 34, 35, 37]. There are 114 lines of records from this source; each line records the names of a nonprofit research center and its hosting university.

3. *The Integrated Postsecondary Education Data System (IPEDS*; https://nces.ed.gov/ipeds/), which provides comprehensive details for U.S. higher education institutions.

4. *Scopus*. Scopus (https://www.scopus.com/) is one of the most comprehensive bibliographic databases of academic articles [38]. From this data source, we retrieved journal articles on

nonprofits and philanthropy authored by scholars from the NME-listed institutions. There are 14,039 lines of records from this data source, each line representing a published article. S1 Appendix details how this scholarship dataset is generated.

5. *American Community Survey* and *Small Area Income and Poverty Estimates* provide social and economic data at county level (i.e., population, poverty rate, and median household income).

To link data from different sources, the first step is to identify and disambiguate the unit of analysis. We use the unique ID from IPEDS (i.e., UNITID) to link and disambiguate universities recorded in different sources. The disambiguation process follows two primary rules. (1) Departments or schools under the same UNITID are combined. The underlying assumption is that different departments under the same UNITID should communicate about their academic and educational activities frequently or share very similar institutional contexts, (2) universities using the same name but with different campuses and different UNITIDs are disambiguated according to their addresses in the NME dataset (e.g., "Indiana University Bloomington" and "Indiana University Indianapolis"). The same rule applies to departments or schools on different campuses.

The IPEDS and ACS datasets are longitudinal, but the master dataset for analysis is an aggregation of historical data: the number of total articles that a university has published on nonprofits and philanthropy. Therefore, we need to decide which year of data we should pull from the IPEDS and ACS. The publication increase has become exponential since the early 2000s according to the distribution of all articles by year and existing studies [5, 18]. We therefore used the year in the middle (i.e., 2010) for the longitudinal datasets in analysis.

Fig 2 shows the simplified schema of the linked datasets (not showing all variables, for brevity's sake).

## Measures

To study the institutional process of knowledge production for practice, the units of analysis in this project were universities. All dependent and independent variables were measures of a university recorded in the NME dataset. To clarify, our study aimed to understand why, *among the U.S. universities with an emphasis on nonprofit management education*, some produced more practical knowledge on nonprofits and philanthropy than others.

**Dependent variable.** The operationalization of the dependent variable, the number of practice-oriented articles published by institutions, relies on coding an article as practice-oriented or not. We adopted a semi-automated approach to code the over six thousand research articles on nonprofits and philanthropy. For a human to code an article as practice-oriented, she identifies informative keywords and contexts from texts and then judges their relevance to practice. Our semi-automated approach simulates this process:

1. Extract topic keywords. We first extracted the keywords of primary topics from all the publications using a state-of-the-art topic-modeling algorithm (detailed in S1 Appendix). The topics are described by keywords extracted from the texts, and the keywords are ranked by their importance to a topic (i.e., the first keyword is the most informative for interpreting the meaning of a topic, etc.).

2. Determine the practice-oriented topics according to existing studies [8, 10]. S1 Appendix has the details.

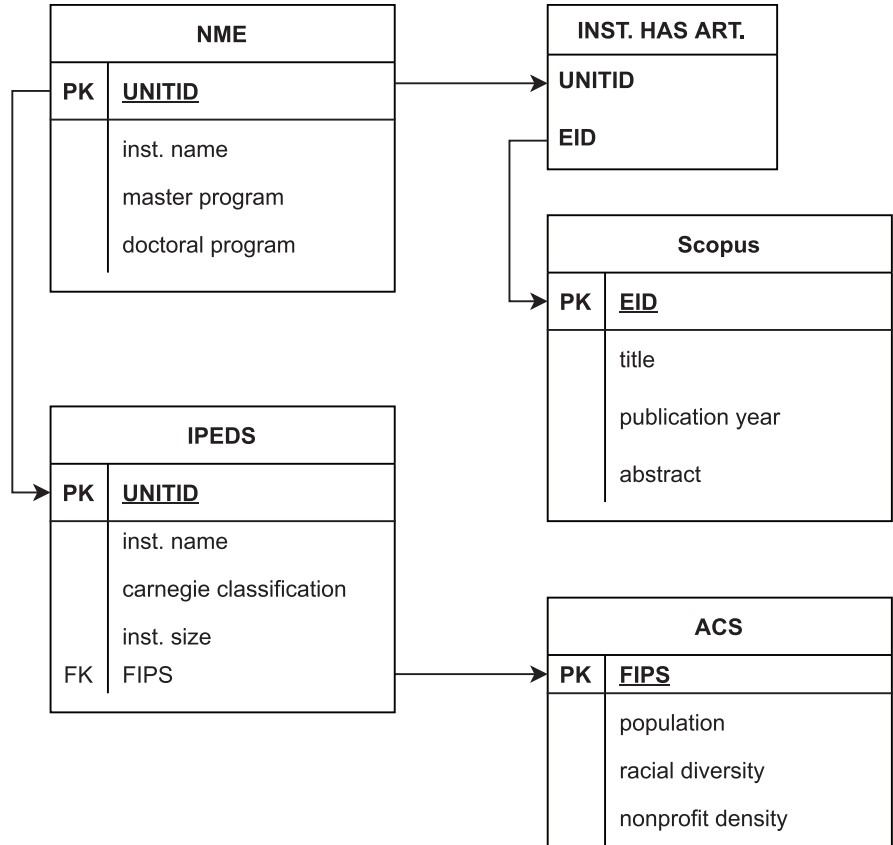

**Fig 2. Simplified schema of linked datasets.** PK = Primary Key, used to uniquely identify the records in a table. FK = Foreign Key, used to link records in different tables. NME = Nonprofit Management Education. IPEDS = The Integrated Postsecondary Education Data System. ACS = American Community Survey. FIPS = Federal Information Processing Standards (County Codes). EID = IDs for uniquely identifying publications. UNITID = IDs for uniquely identifying higher education institutions. INST. HAS ART. = Institution has articles.

3. Assign topics to articles by calculating text similarity. Articles on practice-oriented topics are classified as practice-oriented scholarship. S1 Appendix has the technical details.

4. Validate and improve results. The semi-automated approach to coding may introduce errors. S1 Appendix details how we manually validate the coded results and improve the quality through an iterative process.

Through these steps we can finally calculate the dependent variable: The number of practice-oriented articles published by institutions. While the articles alone may not capture the entirety of practical information available to communities, they are widely recognized as accessible forms of knowledge output and remain an important source of practical information that is feasible to measure within the scope of this study.

**Explanatory variables.** Table 2 lists all the explanatory variables by levels of analysis (variables of interest are underlined). The control variables include: 1) the size of institution, 2) total number of articles on nonprofits and philanthropy (because a university will be likely to have more practice-oriented articles if it produces more publications); 3) basic county demographics pertaining to the location of each university examined in this study, such as total population.

**Table 2. Explanatory variables: Definition, data type, and source.**

| Variable label* | Explanation | Data type | Source** |
|---|---|---|---|
| *Program level* | | | |
| Graduate (+) | Whether a university has graduate-level programs on nonprofits and philanthropy | Binary | NME |
| *Organization level* | | | |
| Research univ. (-) | Whether a university is classified as R1 or R2 in the Carnegie Classification*** | Binary | IPEDS |
| NPS center (+) | Whether a university has a research center on nonprofits and philanthropy | Binary | NRC |
| Inst. size (+) | Institution size, ranked from 1 to 5 according to total students enrolled for credit (including both undergraduate and graduate) | Ordinal | IPEDS |
| #Total articles (+) | Total number of articles on nonprofits and philanthropy | Integer | Scopus |
| *Community level* | | | |
| Poverty rate (+) | Poverty rate by county in which universities are located | Continuous | SAIPE |
| Population (-) | Total population by county in which universities are located | Integer | ACS |

*Variables of interest are underlined. Expected direction of coefficient is in parenthesis.

**See Materials and methods section for the details of the abbreviations.

*** The Carnegie Classification of Institutions of Higher Education is a framework for classifying the U.S. colleges and universities. R1 and R2 indicate "Doctoral Universities—Very high research activity" and "Doctoral Universities—High research activity," respectively. Refer to https://carnegieclassifications.acenet.edu/ for details.

## Results

### An overview of knowledge production for practice

Fig 3 presents the publications and author affiliations by year. The earliest journal article in our research database, titled "Industrial Cooperatives in the Ukrainian S.S.R.," was published by Columbia University in the City of New York in 1951. The number of publications and universities with an NME focus has been steadily increasing since 2000, further strengthening the argument that nonprofit and philanthropic studies is an emerging research field [5, 18, 39]. Publications on practice-oriented topics have increased at a much faster pace than those that are theory-oriented. More than 200 practice-oriented articles were published annually since 2013, and the trend of increase is likely to continue by comparing to the publication trends in the 1990s and 2000s. However, the annual number of publications on theory-oriented topics hovered around 150 since 2013 and did not suggest an upward trend.

The overall production of nonprofit scholarship distributes unevenly. Table 3 lists the details of the top 20 universities by total number of articles (a "university" is an entity with unique ID used by the IPEDS; records from different departments are aggregated by university). Most of these institutions are located in the Midwest and the East. On average, each university published 33.42 (*min* = 1, *max* = 218, *std* = 39.67) articles on nonprofits and philanthropy, and the top 20 most productive universities (8.47% of all the universities; Table 3) published 42.57% (i.e., 2,640 out of 6,201) of all the articles. A geovisualization and details of all the 236 U.S. universities with an NME focus can be accessed here: https://jima.me/us_nme.

The production of practical knowledge also follows an uneven pattern. Although 83.47% of these universities have one or more degree programs with an NME focus at graduate level, most of the nonprofit centers are located in the Midwest and the East. On average, each university published 24.23 (*min* = 1, *max* = 187, *std* = 29.78) practice-oriented articles, and the 20

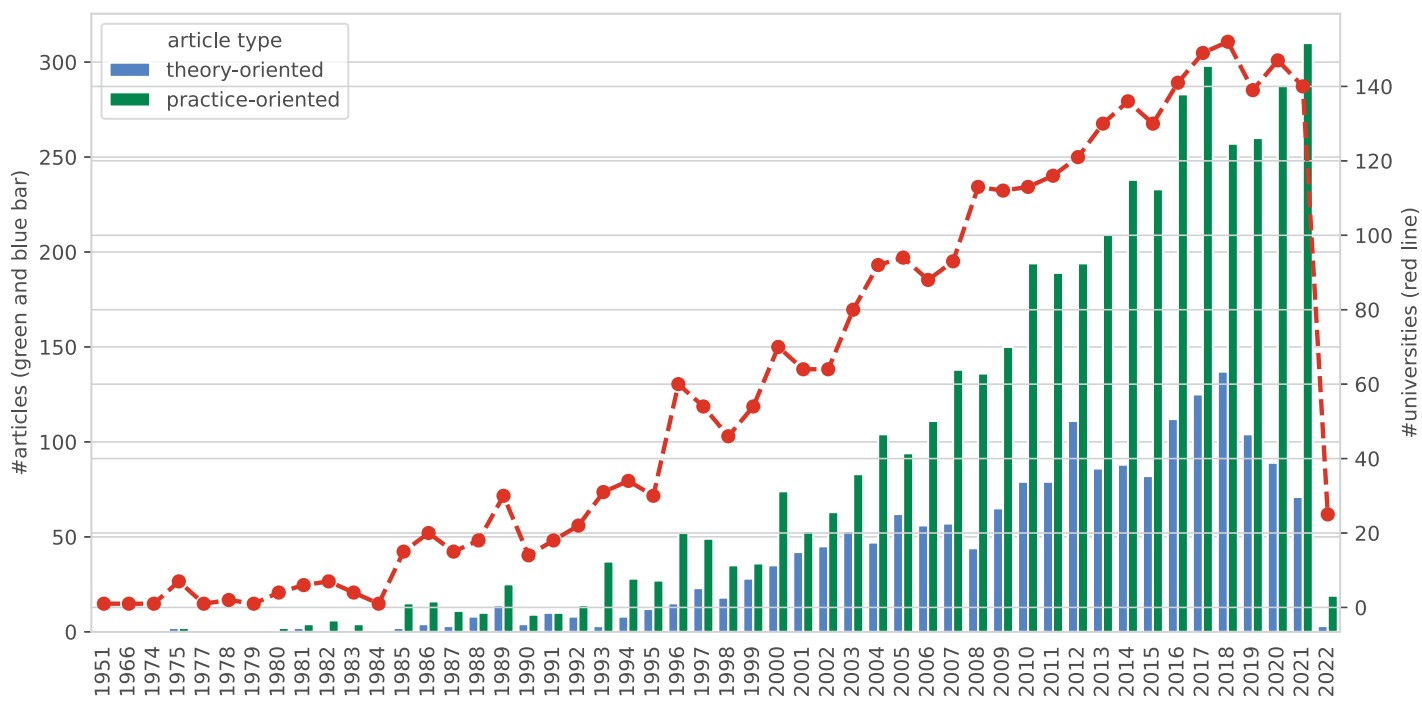

**Fig 3. Distribution of publications and universities by year.** Records for 2022 are incomplete because the research database was compiled in early 2022.

most productive universities published 39.55% of all the practice-oriented articles (1,726 out of 4,364).

In terms of the proportion of practical knowledge, practice-oriented articles constituted 70.38% of all articles (4,364 out of 6,201). At the university level, an average of 76.48% ($min$ = 18.18%, $max$ = 100%, $std$ = 19.28%) of total publications were practice-oriented.

## Major topics

Table 4 lists the major topics extracted from nonprofit scholarship. The topics are represented by the most relevant keywords and ranked by the number of related articles. Among the 24 topics, 15 (62.5%) are practice-oriented. The top three practice-oriented topics focus on the motivation of volunteering (15.72%), fundraising and performance (14.04%), and altruism (10.56%). The top three theory-oriented topics focus on social protest and mobilization (7.54%), deliberation and democracy (5.45%), and regulatory environment in Sub-Saharan Africa (3.44%).

## Predicting knowledge production for practice

Table 5 describes the raw values of the explanatory and dependent variables. S1 Appendix tests the collinearity between explanatory variables and suggests a low statistical risk. All continuous variables are standardized using z-score for the regression analysis.

Table 6 presents the regression results for predicting knowledge production for practice. We build the models stepwise to test the robustness of estimation coefficients. Overall, the models show an exceptional predictive validity. All the models have an adjusted $R^2$ that is larger than 0.90, indicating that 90% of the dependent variable's variance can be explained by the explanatory variables. The most significant contributor to the goodness of fit (i.e., $R^2$) is

**Table 3. Nonprofit scholarship published by U.S. universities with NME focus.**

| UNITID | Inst. Name | Cent. | Grad. | #Art. | #Prac. | %Prac. |
|---:|---|---|---|---:|---:|---:|
| 151111 | Indiana University-Purdue University-Indianapolis | Yes | Yes | 218 | 187 | 85.78% |
| 151351 | Indiana University-Bloomington | No | Yes | 212 | 161 | 75.94% |
| 215062 | University of Pennsylvania | Yes | Yes | 175 | 147 | 84.0% |
| 166027 | Harvard University | No | Yes | 161 | 78 | 48.45% |
| 123961 | University of Southern California | Yes | Yes | 157 | 101 | 64.33% |
| 174066 | University of Minnesota-Twin Cities | Yes | Yes | 156 | 128 | 82.05% |
| 236948 | University of Washington-Seattle Campus | Yes | Yes | 148 | 91 | 61.49% |
| 170976 | University of Michigan-Ann Arbor | No | Yes | 139 | 102 | 73.38% |
| 139959 | University of Georgia | Yes | Yes | 135 | 125 | 92.59% |
| 139940 | Georgia State University | No | Yes | 127 | 108 | 85.04% |
| 215293 | University of Pittsburgh-Pittsburgh Campus | No | Yes | 107 | 61 | 57.01% |
| 130794 | Yale University | No | Yes | 107 | 64 | 59.81% |
| 131469 | George Washington University | No | Yes | 107 | 71 | 66.36% |
| 110635 | University of California-Berkeley | Yes | Yes | 106 | 69 | 65.09% |
| 190150 | Columbia University in the City of New York | No | Yes | 101 | 55 | 54.46% |
| 228723 | Texas A & M University-College Station | Yes | Yes | 100 | 94 | 94.0% |
| 199120 | University of North Carolina at Chapel Hill | No | Yes | 97 | 63 | 64.95% |
| 196413 | Syracuse University | No | Yes | 96 | 84 | 87.5% |
| 199193 | North Carolina State University at Raleigh | Yes | Yes | 96 | 90 | 93.75% |
| 134130 | University of Florida | No | No | 95 | 66 | 69.47% |

*Note*: NME = Nonprofit management education; UNITID = unique ID in the Integrated Postsecondary Education Data System; IUPUI = Indiana University-Purdue University-Indianapolis. Each row shows the following items in order: unique ID in the Integrated Postsecondary Education Data System (ID), instituition name (Inst. Name), whether the institution has a research center on nonprofit (Cent.), whether the institution has a graduate degree program on nonprofits (Grad.), the total number of peer-reviewed journal articles on nonprofits (#Art.), the total number of practice-oriented journal articles (#Prac.), and the percentage of practice-oriented articles (%Prac.).

the number of total articles published by an institution, which makes sense because a university should produce more practice-oriented articles if it has more publications.

**Interpreting the results without considering the interaction effect (i.e., Models 1—5).** At the program level, having a degree program at graduate level has a consistent and significant positive effect on producing practical knowledge (Hypothesis 1). According to Model 5, keeping all other variables equal, universities with graduate programs publish 2.08 more practice-oriented articles than those without (since all continuous variables are transformed using z-score in the regression analysis, the raw number of practice-oriented articles is calculated as $\beta \times StdDV$, where $\beta$ is the coefficient of an explanatory variable, and $StdDV$ is the standard deviation of dependent variable, which is 29.78).

At the organization level, being a research university decreases the production of practical knowledge (Hypothesis 2). According to Model 5, keeping all other variables equal, an R1 or R2 research university publishes 2.38 fewer practice-oriented articles than those not classified as R1 or R2. Having a research center on studying nonprofits and philanthropy, however, can significantly increase knowledge production for practice (Hypothesis 3). According to Model 5, keeping all other variables equal, universities with NPS centers publish 2.68 more practice-oriented articles than those without.

At the community level, the poverty rate has a significant positive impact on producing practical knowledge, suggesting that universities located in disadvantaged counties publish

**Table 4. Topic keywords and the number of corresponding articles.**

|   | Top 5 keywords of topic | Practice-oriented | #Articles | %Articles |
|---|---|---|---|---|
| 0 | motivation, satisfaction, volunteer, inventory, esteem | Yes | 1038 | 15.72% |
| 1 | funder, measurement, performance, grantee, accountability | Yes | 927 | 14.04% |
| 2 | psm, altruism, motivation, motivational, job | Yes | 697 | 10.56% |
| 3 | protest, mobilization, repression, movement, repertoire | No | 498 | 7.54% |
| 4 | deliberative, deliberation, forum, democracy, multilevel | No | 360 | 5.45% |
| 5 | alumnus, college, graduate, advancement, campus | Yes | 242 | 3.67% |
| 6 | elasticity, subsidy, tax, price, charitable | Yes | 241 | 3.65% |
| 7 | venture, entrepreneurs, entrepreneurial, enterprise, founder | Yes | 235 | 3.56% |
| 8 | saharan, sub, africa, regulation, regulatory | No | 227 | 3.44% |
| 9 | diversification, revenue, diversify, earn, portfolio | Yes | 202 | 3.06% |
| 10 | protestant, catholic, religious, church, religion | No | 199 | 3.01% |
| 11 | empathy, advertising, empathic, ad, persuasive | Yes | 196 | 2.97% |
| 12 | ceo, chief, executive, board, director | Yes | 180 | 2.73% |
| 13 | mcb, capitalism, economist, economics, economy | No | 168 | 2.54% |
| 14 | ingos, ingo, hyperlink, nongovernmental, polity | No | 155 | 2.35% |
| 15 | csr, company, corporate, promotional, credibility | Yes | 153 | 2.32% |
| 16 | electoral, election, party, authoritarian, presidential | No | 143 | 2.17% |
| 17 | contracting, contract, contractor, delivery, service | Yes | 90 | 1.36% |
| 18 | katrina, hurricane, disaster, recovery, earthquake | Yes | 57 | 0.86% |
| 19 | twitter, tweet, facebook, media, online | Yes | 56 | 0.85% |
| 20 | cso, aid, reduction, foreign, civil | No | 46 | 0.7% |
| 21 | ukraine, revolution, russian, russia, communist | No | 45 | 0.68% |
| 22 | course, learning, learn, faculty, pedagogical | Yes | 42 | 0.64% |
| 23 | auditor, audit, disclosure, disclose, accounting | Yes | 19 | 0.29% |

*Note*: PSM = Public service motivation [40]. Methods for generating these keywords are detailed in S1 Appendix, "A.2 Semi-automated approach to coding publications."

more practice-oriented articles (Hypothesis 4). According to Model 5, keeping all other variables equal, an increase of 5.12% (i.e., one standard deviation) in a county's poverty rate will increase its universities' practice-oriented articles by 1.40.

**Interpreting the results with interaction effect (i.e., Model 6).** Model 6 in Table 6 can be clarified by Eq 1, which shows how NPS centers mediate the relation between poverty rate and practical knowledge (Hypothesis 5). The model predicts that if a university does not have a research center focusing on studying nonprofits and philanthropy, the poverty rate of the university's county has an insignificant effect on producing practical knowledge. However, universities with NPS centers publish $0.13 \times Pvt$ (*poverty rate*) more articles than those without.

$$TotArt = \begin{cases} .069 \times Grad + (-.072) \times RU + Ctr & \text{if } Center = 0 \\ .069 \times Grad + (-.072) \times RU + .13 \times Pvt + Ctr & \text{if } Center = 1 \end{cases} \tag{1}$$

## Robustness analysis

We tested the statistical robustness of the regression models and the robustness of operationalization. S1 Appendix has the details.

**Table 5. Description of explanatory and dependent variables.**

| Variable | Obs. (%) | Mean (Std) | Min. | 50% | Max. |
|---|---|---|---|---|---|
| *Graduate* | | | | | |
| Yes | 197 (83.47%) | | | | |
| No | 39 (16.53%) | | | | |
| *NPS center* | | | | | |
| Yes | 45 (19.07%) | | | | |
| No | 191 (80.93%) | | | | |
| *Research univ.* | | | | | |
| Yes | 121 (51.27%) | | | | |
| No | 115 (48.73%) | | | | |
| *Inst. size (#total students)* | | | | | |
| 1: Under 1,000 | 2 (0.85%) | | | | |
| 2: 1,000–4,999 | 28 (11.86%) | | | | |
| 3: 5,000–9,999 | 50 (21.19%) | | | | |
| 4: 10,000–19,999 | 65 (27.54%) | | | | |
| 5: 20,000 and above | 91 (38.56%) | | | | |
| *Poverty rate (%)* | 236 (100%) | 16.68 (5.12) | 6.5 | 16.75 | 37.8 |
| *Total articles* | 227 (96.19%) | 33.42 (39.67) | 1 | 19 | 218 |
| *Population* | 236 (100%) | 1,177,800 (1,875,634) | 13,968 | 633,756.5 | 9,974,203 |
| *Prac. article* | 227 (96.19%) | 24.23 (29.78) | 1 | 12 | 187 |

*Note*: NPS = Nonprofit and philanthropic studies. Institution size shows the number of total students enrolled for credit in 2020.

## Discussion

We theorized a useful framework from the perspective of institutional logics to explain why some universities produce more practice-oriented articles in nonprofit studies than others. According to this framework, knowledge production follows the logics of both closed and open systems. A closed-system logic focuses on theory-building in academia, while an open-system logic calls for responding to external environments. For producing practical knowledge, channels that can repackage the needs of open systems to be accepted by closed systems are especially valuable. In testing this theorization, we found that being an R1 or R2 research university (i.e., a closed-system logic) can decrease the number of practice-oriented publications. However, having graduate degree programs with an emphasis on nonprofit management, hosting research centers focusing on nonprofit studies, and being located in disadvantaged communities (i.e., open-system logic) can increase the production of practical knowledge. Research centers on nonprofit studies can serve as channels for logics shifting and cultural repackaging; these centers can mediate the influence of community needs on knowledge production, so that universities with nonprofit research centers become more responsive to solving community issues. The empirical results demonstrate strong robustness across various statistical and research design tests. These findings, along with the theoretical framework proposed, are also valuable references for other interdisciplinary fields of social science.

**Table 6. Predicting knowledge production for practice.**

| | (1) | (2) | (3) | (4) | (5) | (6) |
|---|---|---|---|---|---|---|
| *Program level* | | | | | | |
| Graduate | 0.0548* | 0.0537* | 0.0602* | 0.0602* | 0.0703** | 0.0692** |
| | (1.7) | (1.7) | (1.9) | (1.9) | (2.1) | (2.1) |
| *Organization level* | | | | | | |
| Research univ. | | | -0.0878** | -0.0878** | -0.0800** | -0.0718* |
| | | | (-2.3) | (-2.3) | (-2.1) | (-1.9) |
| NPS center | | 0.0970* | 0.102* | 0.102* | 0.0897* | 0.0766 |
| | | (1.7) | (1.8) | (1.8) | (1.7) | (1.4) |
| *Community level* | | | | | | |
| Poverty rate | | | | | 0.0468** | 0.0146 |
| | | | | | (2.4) | (1.0) |
| *Shifting logics* | | | | | | |
| Center × Poverty | | | | | | 0.128*** |
| | | | | | | (2.8) |
| *Controls* | | | | | | |
| Inst. size | -0.00236 | 0.000904 | 0.0218 | 0.0218 | 0.0123 | 0.0153 |
| | (-0.1) | (0.1) | (1.3) | (1.3) | (0.7) | (0.9) |
| #Total articles | 0.974*** | 0.959*** | 0.970*** | 0.970*** | 0.970*** | 0.960*** |
| | (21.3) | (20.3) | (20.0) | (20.0) | (21.0) | (21.0) |
| Population | -0.0189 | -0.0225 | -0.0207 | -0.0207 | -0.0151 | -0.0138 |
| | (-1.4) | (-1.5) | (-1.4) | (-1.4) | (-1.1) | (-1.1) |
| Observations | 220 | 220 | 220 | 220 | 220 | 220 |
| Adjusted $R^2$ | 0.937 | 0.938 | 0.938 | 0.938 | 0.940 | 0.943 |

*Note*: DV = number of practice-oriented articles. All continuous variables are normalized using z-score. $t$ statistics in parentheses.

*$p < 0.10$,

**$p < 0.05$,

***$p < .01$

## Practical implications of the framework

The theoretical framework of this study provides compelling information about factors associated with the production of practice-oriented knowledge in nonprofit studies across different universities. In general, efforts with an open-system logic are key to producing more practical knowledge. This study empirically confirms the effectiveness of two types of such efforts within universities: graduate programs with an emphasis on nonprofit management, and research centers on nonprofit studies.

More initiatives with an open-system logic can be pursued within universities: service-learning curricula, continuing education programs, and outreach newsletters, for example. However, to be culturally accepted by universities, these efforts have to be repackaged. As Lounsbury and Pollack suggested in their study of service learning in U.S. higher education, the efforts aimed at bringing novel values into universities also must be recognized under existing conventions, which usually carry a closed-system logic [32].

Channels that can shift the logics and repackage the cultures are key to bridging open and closed systems. Numerous qualitative studies suggest that research centers on nonprofit studies can serve this role because a central rationale of such centers is to facilitate exchange between scholars and practitioners [17, 34, 36]. Scholars further suggested that connecting universities with local communities and providing entrepreneurial spaces for faculty members are

two primary functions of these research centers [35, pp. 15–17]. This study empirically confirms these observations: for universities with NPS research centers, external environment has a more substantial effect on these institutions' knowledge production for practice.

Researchers also noted that traditional university structures have absorbed several nonprofit research centers in the past decade, suggesting that such a trend may represent either the successful institutionalization of this research field or the centers' loss of independence [35, p. 20]. Based on our analysis, we lean to the positive side: it is beneficial for the nonprofit research centers to step closer to the closed system so that they can better serve their logic-shifting and cultural-repackaging roles.

## Matching research priorities between scholars and practitioners

Even when scholars' and practitioners' interests substantially overlap, their research priorities can differ substantially [8]. For example, the management of volunteers is among practitioners' highest priorities, but that subject is marginalized on nonprofit scholars' research agenda [8, p. 301]. The rankings of research topics in Table 4 are almost consistent with those surveyed in the early 1990s [8], while practitioners' needs may already have changed, and they may have more interest in newer topics such as the application of information technologies, which is absent from our list, and online social networks, which is ranked only 20th out of 23 on our list.

Practical needs are only one of the many factors that determine a scholar's research priorities. Matured research paradigms may carry more weight in deciding research agenda [41]. Therefore, research topics with established theoretical frameworks can generate publications more easily than those without—the so-called "puzzle-solving" process in knowledge production. Table 4 supports this point. Those researching the most productive topics can build their research primarily on paradigms from psychology, finance, and management—the areas where more research paradigms are shared and consensus level is high [42]. For research topics with less established theoretical paradigms, such as social enterprise, fewer articles are published.

Therefore, knowledge production for practice should be examined in terms not only of volume but also of priorities. Practitioners' pressing concerns must be reflected and prioritized in timely fashion on scholars' agendas, and institutions for shifting logics are essential channels for this task.

## Limitations and future directions

This study uses institutional logics as its primary theoretical lens. Therefore, it focuses on institutional factors and external environments rather than on individuals. Future studies can explore how individual-level factors, such as personality traits and social networks, can influence the knowledge production process.

Regarding the research design of this study, a major limitation is the operationalization of "practical knowledge." First, because we use research topics as the instrument for coding article abstracts, it is possible that an article labeled as practice-oriented is a theory-building piece. Moreover, because all the research topics are extracted from academic scholarship, they may reflect the research interests only of scholars, not practitioners.

For future projects, the application of new technologies such as machine learning and cross-language modeling offers promising possibilities [43]. According to our assessment, scholarship on practical topics still predominates in terms of volume in our research field. However, we have limited information to evaluate the practical impact of these articles. Data sources tracking, for example, citations in policy documents, could be used to create direct

outcome measures. Furthermore, practitioners may rarely pick up academic scholarship directly. The publishers' paywall is one reason, but a more important reason may be the lack of a database platform where practitioners can look up scholarship according to their interests and communicate their priorities. Such a platform could be an important venue for sharing knowledge between practitioners and scholars.

Lastly, while this study offers useful insights into some of the factors that influence the production of practice knowledge, it is crucial to recognize that there are likely additional factors not addressed in the current paper. While the framework presented is grounded in empirical evidence, it should be considered as a preliminary foundation rather than a definitive conclusion. Future endeavors should incorporate a broader range of nuances and complexities when investigating the process of generating practical knowledge.

## Supporting information

**S1 Appendix. Methods and robustness analysis.**
(PDF)

## Acknowledgments

We thank Roseanne M. Mirabella and Benjamin J. Lough for generously sharing their datasets and insights. We thank Brenda K. Bushouse, Huafang Li, Peter Weber, and Sydney Wilburn for their kind and constructive comments. We thank the attendees of the 2022 ARNOVA Annual Conference and the cloud computing resources through the Texas Advanced Computing Center at UT Austin [44]. We thank Kate Hartford for editing and proofreading.

## Author Contributions

**Conceptualization:** Ji Ma.

**Data curation:** Ji Ma, Joycelyn Ovalle, Yan Wang.

**Formal analysis:** Ji Ma.

**Funding acquisition:** Ji Ma.

**Investigation:** Ji Ma.

**Methodology:** Ji Ma, Yan Wang.

**Project administration:** Ji Ma, Joycelyn Ovalle.

**Resources:** Ji Ma.

**Software:** Ji Ma.

**Supervision:** Ji Ma.

**Validation:** Ji Ma, Yan Wang.

**Visualization:** Ji Ma.

**Writing – original draft:** Ji Ma.

**Writing – review & editing:** Ji Ma, Joycelyn Ovalle.

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
