## [Decision Letter · Decision Letter 0]

6 Jun 2023

PONE-D-23-08626Institutional factors influencing knowledge production for practice: Evidence from nonprofit studiesPLOS ONE

Dear Dr. Ma,

Thank you for submitting your manuscript to PLOS ONE. After careful consideration, we feel that it has merit but does not fully meet PLOS ONE’s publication criteria as it currently stands. Therefore, we invite you to submit a revised version of the manuscript that addresses the points raised during the review process.

We look forward to receiving your revised manuscript.

Kind regards,

Claudia Noemi González Brambila, Ph.D.

Academic Editor

PLOS ONE

“The project is partly funded or supported by (1) the Academic Development Funds from the RGK Center, (2) a 2021-22 PRI Award from the LBJ School, (3) library resources through the IU Lilly Family School of Philanthropy, and (4) cloud computing resources through the Texas Advanced Computing Center at UT Austin.”

3. We note that Figure 4 and Figure C1 in your submission contain [map/satellite] images which may be copyrighted. All PLOS content is published under the Creative Commons Attribution License (CC BY 4.0), which means that the manuscript, images, and Supporting Information files will be freely available online, and any third party is permitted to access, download, copy, distribute, and use these materials in any way, even commercially, with proper attribution. For these reasons, we cannot publish previously copyrighted maps or satellite images created using proprietary data, such as Google software (Google Maps, Street View, and Earth). For more information, see our copyright guidelines: http://journals.plos.org/plosone/s/licenses-and-copyright.

a. You may seek permission from the original copyright holder of Figure 4 and Figure C1 to publish the content specifically under the CC BY 4.0 license. 

Reviewers' comments:

Reviewer's Responses to Questions

**Comments to the Author**

1. Is the manuscript technically sound, and do the data support the conclusions?

Reviewer #1: Yes

Reviewer #2: Yes

2. Has the statistical analysis been performed appropriately and rigorously? 

Reviewer #1: Yes

Reviewer #2: Yes

3. Have the authors made all data underlying the findings in their manuscript fully available?

Reviewer #1: Yes

Reviewer #2: Yes

4. Is the manuscript presented in an intelligible fashion and written in standard English?

Reviewer #1: Yes

Reviewer #2: Yes

5. Review Comments to the Author

Reviewer #1: This study tries and test theories to explain and understand why some

universities produce more practical knowledge than others using nonprofit studies as

an example. Knowledge production

follows the logics of both closed and open systems. A closed-system logic focuses on 361

theory-building in academia, while an open-system logic calls for responding to external 362

environments

Reviewer #2: This review is for the manuscript titled "Institutional factors influencing knowledge production for practice: Evidence from nonprofit studies." The premise of this manuscript is to produce evidence (as operationalized by published manuscripts) of factors that are associated with production of practice-based articles. The authors are applauded for their efforts to operationalize strategies to bridge the science-to-service gap. They used a thoughtful approach to determine what factors are most important.

This manuscript has the potential to make a substantial contribution to the field and, as such, I have a few recommendations to address where I see some gaps and needs in the overall manuscript.

Small point, there’s a typo on page 2, second line “studies” should read “studied”.

Table 1. Expand Poverty rate to say “Poverty rate at institutional location” You haven’t yet introduced how you conceptualize poverty rate when the table is introduced, so it’s confusing to the reader who might interpret community-based institutions as increasing the poverty rate.

The introduction is entirely missing any discussion of implementation science, a field that was likewise developed to bridge the science-practice/practice-science gap. IS principles could further enrich the introduction, particularly discussions about the importance of context, co-creation, team science, etc.

In the measures section, it strikes me as a little odd that the authors critiqued the limitations of articles, yet are using them as the key dependent variable. Given this, the authors might explain a bit more why they chose this measure for the dependent variable and if they considered other sources of information that provide practical information to communities.

Under explanatory variables (page 8), clarify if the basic county demographics are of the institution or of the subjects of the article.

Likewise, in Table 2, specify that poverty rate by county is where the institution resides, not the study.

Figure 4, it is unclear why University of Hawaii at Manoa was not included due to space. I would recommend including it for completeness.

I struggled a bit with understanding Table 4. It might be more understandable if the keywords were listed in two tables, one for practice-oriented and one for theory-oriented, and remove the underscores. As it is, it's not clear how to interpret this table, even with the explanation in the text. It’s also not entirely clear how the combination of keywords were derived. More information is needed.

Table 6. Is there a way to put the odds ratios in the table? It would make understanding the narrative a bit easier to follow. Also, are you sure the Adjusted R2 are the same across all six models? This may be a typo.

Discussion section. The first sentence in Practical implications of the framework you state “The theoretical framework coherently explains knowledge production for practice in nonprofit studies and has the potential to guide future action.” This seems to be a bit of an overreach given variables that were not able to be included in your model. I would recommend a softer frame for the results “The Theoretical framework provides compelling information about factors associated with…”

Limitations – I think it is important to acknowledge there are likely more factors than the ones able to be distilled in this manuscript that influence the likelihood of practice-based manuscripts.

6. PLOS authors have the option to publish the peer review history of their article (what does this mean?). If published, this will include your full peer review and any attached files.

---

## [Author Response · Author response to Decision Letter 0]

31 Jul 2023

Please see the response letter attached.

---

## [Decision Letter · Decision Letter 1]

18 Sep 2023

PONE-D-23-08626R1Institutional factors influencing knowledge production for practice: Evidence from nonprofit studiesPLOS ONE

Dear Dr. Ma,

Thank you for submitting your manuscript to PLOS ONE. After careful consideration, we feel that it has merit but does not fully meet PLOS ONE’s publication criteria as it currently stands. Therefore, we invite you to submit a revised version of the manuscript that addresses the points raised during the review process.

We look forward to receiving your revised manuscript.

Kind regards,

Claudia Noemi González Brambila, Ph.D.

Academic Editor

PLOS ONE

Journal Requirements:

Reviewers' comments:

Reviewer's Responses to Questions

**Comments to the Author**

1. If the authors have adequately addressed your comments raised in a previous round of review and you feel that this manuscript is now acceptable for publication, you may indicate that here to bypass the “Comments to the Author” section, enter your conflict of interest statement in the “Confidential to Editor” section, and submit your "Accept" recommendation.

Reviewer #2: All comments have been addressed

2. Is the manuscript technically sound, and do the data support the conclusions?

Reviewer #2: Yes

3. Has the statistical analysis been performed appropriately and rigorously? 

Reviewer #2: Yes

4. Have the authors made all data underlying the findings in their manuscript fully available?

Reviewer #2: Yes

5. Is the manuscript presented in an intelligible fashion and written in standard English?

Reviewer #2: Yes

6. Review Comments to the Author

Reviewer #2: Overall, the authors did a laudable job addressing feedback from the prior review. There remain only a couple of small areas to address:

1) Page 7, paragraph starting on line 240, carefully proofread for tense changes

2) Consider ordering Table 3 by % or # of articles

3) Table 5, note if institutional size is inclusive of graduate and undergraduate, or graduate only.

4) The limitations section was robust and much improved. I think this was implied in the section, but you may want to consider further highlighting that you were only able to look at published manuscripts and did not look at reports or other products that might have been produced that were practice-oriented.

7. PLOS authors have the option to publish the peer review history of their article (what does this mean?). If published, this will include your full peer review and any attached files.

Reviewer #2: No

---

## [Author Response · Author response to Decision Letter 1]

18 Sep 2023

Please refer to the attached files.

---

## [Decision Letter · Decision Letter 2]

11 Oct 2023

Institutional factors influencing knowledge production for practice: Evidence from nonprofit studies

PONE-D-23-08626R2

Dear Dr. Ma,

We’re pleased to inform you that your manuscript has been judged scientifically suitable for publication and will be formally accepted for publication once it meets all outstanding technical requirements.

Kind regards,

Claudia Noemi González Brambila, Ph.D.

Academic Editor

PLOS ONE

Additional Editor Comments (optional):

Reviewers' comments:

Reviewer's Responses to Questions

**Comments to the Author**

1. If the authors have adequately addressed your comments raised in a previous round of review and you feel that this manuscript is now acceptable for publication, you may indicate that here to bypass the “Comments to the Author” section, enter your conflict of interest statement in the “Confidential to Editor” section, and submit your "Accept" recommendation.

Reviewer #2: All comments have been addressed

2. Is the manuscript technically sound, and do the data support the conclusions?

Reviewer #2: Yes

3. Has the statistical analysis been performed appropriately and rigorously? 

Reviewer #2: Yes

4. Have the authors made all data underlying the findings in their manuscript fully available?

Reviewer #2: Yes

5. Is the manuscript presented in an intelligible fashion and written in standard English?

Reviewer #2: Yes

6. Review Comments to the Author

Reviewer #2: Thank you for careful consideration of peer-review recommendations. All comments appear sufficiently addressed.

7. PLOS authors have the option to publish the peer review history of their article (what does this mean?). If published, this will include your full peer review and any attached files.

---

## [Editor Report · Acceptance letter]

16 Oct 2023

PONE-D-23-08626R2 

Institutional factors influencing knowledge production for practice: Evidence from nonprofit studies 

Dear Dr. Ma:

I'm pleased to inform you that your manuscript has been deemed suitable for publication in PLOS ONE. Congratulations! Your manuscript is now with our production department. 

Kind regards, 

on behalf of

Dr. Claudia Noemi González Brambila 

Academic Editor

PLOS ONE